# Recent Advances in CoSe$_x$ and CoTe$_x$ Anodes for Alkali-ion Batteries

**Yuqi Zhang, Zhonghui Sun \*, Dongyang Qu \*, Dongxue Han and Li Niu**

College of Chemistry and Chemical Engineering, Guangzhou University, Guangzhou 510006, China; 2112105042@e.gzhu.edu.cn (Y.Z.); dxhan@gzhu.edu.cn (D.H.); lniu@gzhu.edu.cn (L.N.)
\* Correspondence: cczhsun@gzhu.edu.cn (Z.S.); dyqu@gzhu.edu.cn (D.Q.)

**Abstract:** Transition metal selenides have narrow or zero band-gap characteristics and high theoretical specific capacity. Among them, cobalt selenide and cobalt telluride have some typical problems such as large volume changes, low conductivity, and poor structural stability, but they have become a research hotspot in the field of energy storage and conversion because of their high capacity and high designability. Some of the innovative synthesis, doping, and nanostructure design strategies for CoSe$_x$ and CoTe$_x$, such as CoSe-InCo-InSe bimetallic bi-heterogeneous interfaces, CoTe anchoring MXenes, etc., show great promise. In this paper, the research progress on the multistep transformation mechanisms of CoSe$_x$ and CoTe$_x$ is summarized, along with advanced structural design and modification methods such as defect engineering and compositing with MXenes. It is hoped that this review will provide a glimpse into the development of CoSe$_x$ and CoTe$_x$ anodes for alkali-ion batteries.

**Keywords:** CoTe$_x$; CoSe$_x$; anodes; alkali-metal-ion batteries

## 1. Introduction

The demand for clean renewable energy sources such as wind and solar power has increased due to the energy problems and environmental pollution from traditional fossil energy. Lithium-ion batteries (LIBs) were born as high-performance energy storage devices to compensate for the inconvenience of storing clean energy [1,2]. Today, lithium resources are increasingly scarce; thus, sodium-ion batteries (SIBs) and potassium-ion batteries (PIBs) have become two of the hopes to solve the problem of resource shortage [3,4]. The prerequisite for further application of sodium/potassium-ion batteries is to prepare negative electrode materials with appropriate working voltage, excellent cycling performance, and high energy density. Anode materials can be divided into three types: intercalation type (e.g., carbon [5] and TiO$_2$ [6]), alloy type (Si, Ge, Sb [7], etc.), and conversion type (Co$_3$O$_4$ [8], Fe$_x$S [9], etc.). Although an intercalated anode has a stable structure and good electrochemical cycling stability, it still has the disadvantages of low energy density and limited specific capacity. In addition, the alloy-type anode has a high capacity, but the large volume change caused by the alloy reaction brings the disadvantage of poor stability. Compared with the former two, the conversion negative electrode has a reasonably high specific capacity and a small volume expansion, making it a potential choice for ion batteries' anodes [10].

In recent years, transition metal chalcogenides (TMCs) have been widely studied as representative conversion-type anode materials due to their stable performance, high theoretical specific capacity, abundant natural resources, diverse types, and low cost. They hold great potential in meeting the demand for high-energy-density anodes in alkali-ion batteries [11]. Compared to transition metal sulfides, transition metal selenides exhibit higher volumetric capacity as anionic battery anodes. Additionally, the weaker M-Se bond compared to M-O and M-S bonds improves the reaction kinetics of transition metal selenides, resulting in higher energy density and electronic conductivity. Moreover, transition metal selenides have larger interlayer spacing, which helps alleviate volume expansion issues during cycling, especially for larger alkali metal ions such as sodium and potassium.



Therefore, they possess advantages in electrochemical energy storage applications [12–17]. Compared to transition metal oxides and sulfides, transition metal tellurides possess lower electronegativity, the highest conductivity, and the highest volumetric capacity. They are typical layered materials with a large interlayer space, which is beneficial for rapid ion transport in electrodes. This results in excellent electrode wetting and ion diffusion kinetics. So far, many selenides (such as $CoSe_x$ [18], $ZnSe_x$ [19], $MoSe_x$ [20], and $FeSe_x$ [21]) and tellurides (e.g., $NiTe_x$ [22], $FeTe_x$ [23]) have been studied as ion battery anode materials.

Cobalt-based materials have become important materials in various fields (such as hydrodesulfurization and hydrodefluorination) due to their unique catalytic activity, electrochemical performance, and ferromagnetism [24–26]. Among them, $LiCoO_2$ is one of the most successful commercial cathode materials for lithium-ion batteries (LIBs) [27]. Based on the success of $LiCoO_2$, research on $Na_xCoO_2$ as a cathode material for sodium-ion batteries has also been widely conducted [28]. In terms of anode materials, cobalt has various oxidation states, and cobalt-based chalcogenides have diverse types and high theoretical sodium storage capacity, making them highly favored by researchers. Cobalt-based composite anodes, such as cobalt oxide [29], cobalt phosphide, and cobalt–aluminum alloys, have theoretical capacities of 700~1000 mAh g$^{-1}$, which is 2~3 times that of graphite anodes. They have diverse crystal structures and low cost, receiving increasing attention in recent years. Among them, $CoSe_x$ is widely used as an anode material for alkali-ion batteries due to its large capacity, strong conductivity, and weak electronegativity [18,30–34]. $CoTe_x$ is considered to be a potential choice with excellent electrochemical performance in terms of metallicity, magnetism, electronics, and electrocatalysis [35]. In the past few years, extensive research has shown that cobalt selenides and tellurides can be used as potential anodes with high capacity or improved rate performance. However, their inherent volume changes as conversion anodes, relatively low electronic conductivity, and side reactions can lead to significant reversible capacity loss and low active material utilization. Recently, various energy storage batteries have been working on mitigating these issues and have achieved successes, such as forming hybrid/composite samples to enhance conductivity, optimizing electrode design and electrolyte content, and adjusting voltage windows [31].

At present, some progress has been made in the research of energy storage and conversion of transition metal selenides and tellurides—typically cobalt-based compounds [36]. However, it is necessary to study the application and reaction mechanisms of cobalt selenide and cobalt telluride in the three types of metal-ion battery in a timely and systematic manner. This paper reviews the recent progress in the preparation and application of $CoSe_x$/$CoTe_x$-based electrodes (Figure 1). The electrode design and reaction mechanism are introduced in detail. Finally, the challenges and opportunities in this field are presented (Figure 2). This review will significantly advance the research on innovative $CoSe_x$/$CoTe_x$ design to improve its application in energy storage devices.

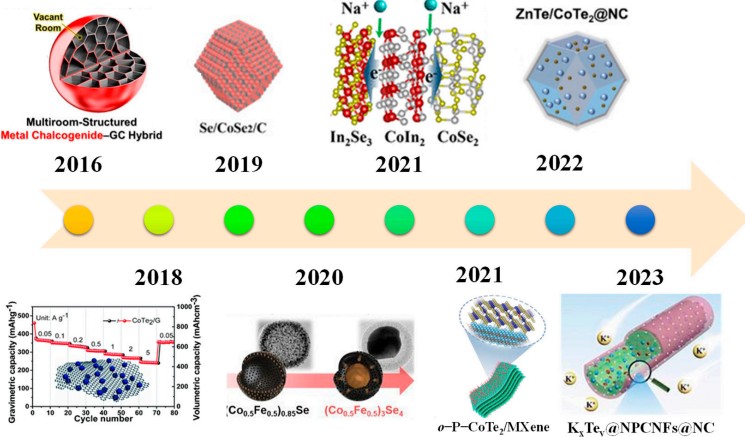

**Figure 1.** Typical recent advances in the synthesis of $CoSe_x$/$CoTe_x$–based electrodes, Refs. [37–45].

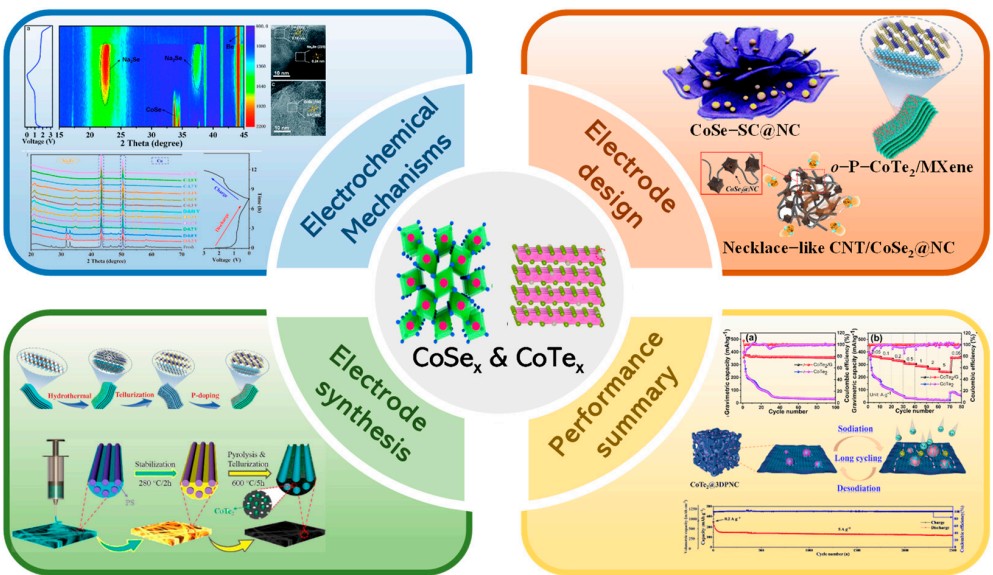

**Figure 2.** This paper summarizes from four perspectives, including electrochemical mechanisms, electrode design, synthesis, and performance; Refs. [38,42,46–48].

## 2. Electrochemical Reaction Mechanisms

As anode materials during charging and discharging processes, different metal-based compounds usually have different kinds of reactions, such as intercalation, conversion, and alloying. For example, intercalation–conversion–alloying reactions take place during the charging and discharging of Sn- and Sb-based compounds; Mo-, W-, Fe-, and Ni-based compounds mainly have intercalation and conversion reactions, while Nb-based compounds only have intercalation reactions [49]. The metal-based anode affects its reaction type. TMCs with conversion reactions can accommodate more alkali ions through the multi-electron transfer process and have a higher capacity. In addition, the combination with chalcogenide elements is conducive to the conversion reaction during the charge–discharge process. Due to the weak TM–B (B = Se and Te) bonds, TMBs have a high theoretical capacity, as well as excellent electrical conductivity and convenient ion transport channels. $Co_xB_y$ has been used as a potential anode in alkali-ion batteries. The storage mechanism between $Co_xB_y$ in LIBs/SIBs/PIBs is the conversion reaction that eventually generates Co and $A_xB$ (A = Li, Na, K). In particular, a few intermediate reactions may occur during the $Co_xB_y$ discharge processes, as shown in the following equations:

$$Co_xB_y + A^+ + e^- \rightarrow ACo_xB_y \tag{1}$$

$$Co_xB_y + A^+ + e^- \rightarrow Co_xB_{y-1} + AB \tag{2}$$

Deciphering the electrochemical reaction mechanism of $CoSe_x$ and $CoTe_x$ is significant to enhance their application in batteries. The latest progress on the reaction mechanisms of cobalt selenide and cobalt telluride for the anode material of LIBs, SIBs, and PIBs is summarized below.

### 2.1. CoSe_x

Many types of research on the storage mechanisms of $CoSe_x$ anodes have been conducted. Among them, the storage path of $CoSe_x$ in PIBs is different from that in SIBs and LIBs. Yu et al. [34] studied the potassium-ion storage mechanism of octahedral $CoSe_2$ nanotubes threaded with N-doped carbon through first-principles calculation (Figure 3a–f). The calculation showed that the formation energies of $CoK_xSe_x$ were all positive, indicating that the insertion of $K^+$ into $CoSe_2$ is difficult to achieve, and it is more inclined to replace Co to generate $K_2Se$. The reason for this is that potassium atoms are too large to penetrate

the lattice of CoSe$_2$, whereas the non-stoichiometric selenide CoSe$_{1-x}$ (Co$_3$Se$_4$) has a wider lattice than CoSe$_2$, achieving K$^+$ embedding. The actual reaction process can be explained as follows:

$$CoSe_2 + \frac{4}{3}K^+ + \frac{4}{3}e^- \rightarrow \frac{1}{3}Co_3KSe_4 \tag{3}$$

$$Co_3KSe_4 + K^+ + e^- \rightarrow 3CoSe + K_2Se \tag{4}$$

$$CoSe + 2K^+ + 2e^- \rightarrow Co + K_2Se \tag{5}$$

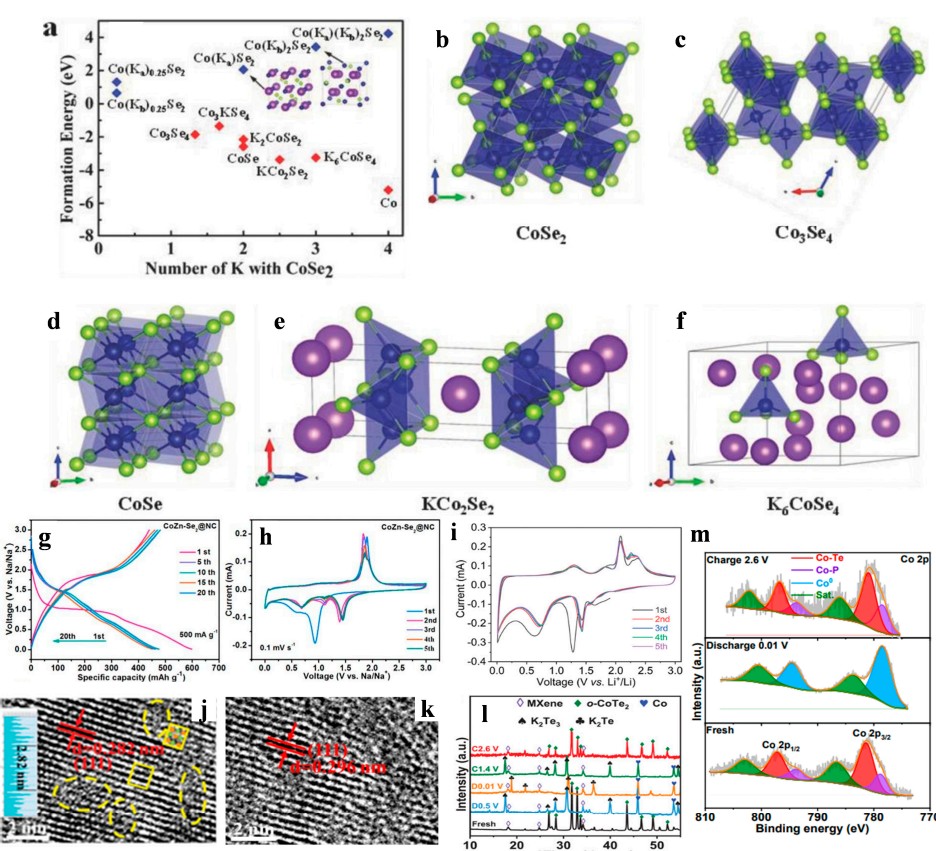

**Figure 3.** (**a**) The formation energy of conversion products. (**b**–**f**) Schematic molecular structures using in this work; Ref. [34]. (**g**) The cycling performance and (**h**) CV curve; Ref. [50]. (**i**) CoSe/G CV curve; Ref. [51]. (**j**) HRTEM for o–P–CoTe$_2$/MXene and (**k**) after discharging to 1.0 V. (**l**) Ex situ XRD and (**m**) ex situ XPS Co 2p; Ref. [38].

In contrast to PIBs, Jiang et al. [50] found the appearance of three new peaks in cyclic voltammetry (CV) curves (Figure 3h), indicating that CoSe$_2$ turns into Na$_x$CoSe$_2$, CoSe, and Co. The oxidation peaks at 1.53 V and 1.89 V represent the intermediate product Na$_x$CoSe$_2$ and CoSe$_2$, which is the fully charged product during the charge process, respectively. The locations of these peaks are highly consistent throughout the five cycles of reduction and oxidation, together with the corresponding position of the charging and discharging platform (Figure 3g), testifying to the reversibility of the charge–discharge process. The actual reaction mechanism can be summarized as follows:

$$CoSe_2 + xNa^+ + xe^- \leftrightarrow Na_xCoSe_2 \tag{6}$$

$$Na_xCoSe_2 + (2-x)Na^+ + (2-x)e^- \leftrightarrow CoSe + Na_2Se \tag{7}$$

$$CoSe + 2Na^+ + 2e^- \leftrightarrow Na_2Se + Co \tag{8}$$

$$2Na_2Se + Co \leftrightarrow CoSe_2 + 4Na^+ + 4e^- \tag{9}$$

In LIBs, Jiang et al. [51] found three prominent peaks of CoSe/G at 1.42, 1.27, and 0.63 V in the first cathodic scan through the first five CV curves in LIBs (Figure 3i). Li$^+$ inserted into the CoSe showed a prominent peak at 1.42 V, corresponding to the formation of Li$_x$CoSe. The solid–electrolyte interphase (SEI) formed at 1.27 V. The wide peak at 0.63 V can be ascribed to the conversion reaction from Li$_x$CoSe to Li$_2$Se and Co. The anode peaks at 1.28 and 2.1 V signify the oxidation reactions that convert Li$_2$Se and Co into CoSe. The reaction process of CoSe anode in LIBs can be expressed as follows:

$$CoSe_2 + xLi^+ + xe^- \leftrightarrow Li_xCoSe \tag{10}$$

$$Li_2CoSe + (2-x)Li^+ + 2e^- \leftrightarrow Co + Li_2Se \tag{11}$$

*2.2. CoTe$_x$*

As an emerging potential anode, the storage mechanism of CoTe$_x$ is being extensively studied. Xu et al. [38] performed high-resolution transmission electron microscopy (HRTEM), X-ray diffraction (XRD), and X-ray photoelectron spectroscopy (XPS) analyses on CoTe$_2$ nanowires with a Te vacancy (o-P-CoTe$_2$/MXene) in different discharge and charge states to study the potassium storage behavior. When the half-cell discharged to 1.0 V, the HRTEM (Figure 3j) showed that the lattice spacing of the (111) plane expanded significantly to 0.296 nm compared with the initial state (Figure 3k, 0.282 nm), which proves the existence of K$^+$ intercalation above 1.0 V. At 0.5 V, peaks of K$_2$Te$_3$ and Co were observed in the ex situ XRD spectra (Figure 3l). This observation confirmed the conversion reaction from CoTe$_2$ to Co and K$_2$Te$_3$. After further discharging to 0.01 V, a peak of K$_2$Te was observed, indicating that K$_2$Te$_3$ was further converted into the final reduction product (K$_2$Te). When the half-cell was charged to 1.4 V, K$_2$Te converted to K$_2$Te$_3$ and Co. When the anode was further charged to 2.6 V, the peak of o-P-CoTe$_2$/MXene reappeared. Ex situ XPS (Figure 3m) spectra further confirmed that the peak of Co 2p$^{3/2}$ continued to shift from 781.4 eV to 778.4 eV during complete discharge, corresponding to the formation of Co. Therefore, the o-P-CoTe$_2$/MXene anode conversion mechanism in PIBs can be described by the following reactions:

$$CoTe_2 + xK^+ + xe^- \leftrightarrow K_xCoTe_2 \tag{12}$$

$$3K_xCoTe_2 + (4-3x)K^+ + (4-3x)e^- \leftrightarrow 3Co + 2K_2Te_3 \tag{13}$$

$$K_2Te_3 + 4K^+ + 4e^- \leftrightarrow 3K_2Te \tag{14}$$

Recently, Li et al. [41] published a more detailed supplement on the storage mechanism of CoTe$_x$ in PIBs. As shown in the in situ XRD pattern (Figure 4a), after discharging to 1.21 V, the peak of CoTe$_2$ shifted to a lower angle, indicating that K$^+$ was embedded into the CoTe$_2$ lattice and K$_x$CoTe$_2$ formed. Upon further discharging to 0.01 V, the peak of CoTe$_2$ became weaker, corresponding to the conversion of CoTe$_2$ to Co and K$_x$Te$_y$. When charged to 3 V, the peak of CoTe$_2$ reappeared. In situ XPS analysis (Figure 4b) showed that the metallic Co phase appeared in the high-resolution Co 2p spectrum when the cell was discharged to 0.8 V. In the Te 3d spectrum, K$_5$Te$_3$ appeared, indicating that K$_x$CoTe$_2$ was converted to K$_5$Te$_3$ at 0.8 V. When discharging to 0.4 V, the cobalt peak strengthened and a new peak of K$_2$Te appeared in the Te 3d spectrum. Upon further discharging to 0.1 V, K$_5$Te$_3$ was almost transformed into K$_2$Te. The electrochemical reaction mechanism was further studied by transmission electron microscope (TEM), energy-dispersive X-ray spectroscopy (EDS), and HRTEM (Figure 4c–h). The TEM and EDS patterns proved that the structure remains stable and the elements are evenly distributed after the cycle. The

discharge HRTEM image shows the (100) crystal plane of Co, the (251) crystal face of $K_5Te_3$, and the (311) and (200) crystal faces of $K_2Te$ (Figure 4d). In addition, upon full charge, the (011) plane of $CoTe_2$ (Figure 4g) appears, which is consistent with the in situ XRD results. In summary, the above results show that $K_xCoTe_2$ converted totally to $K_5Te_3$, and then $K_5Te_3$ converted to $K_2Te$ during the discharge process, and the authors clearly express the evolution of polytellurides in Figure 4i. Therefore, the relevant reversible reactions are summarized below:

$$CoTe_2 + xK^+ + xe^- \rightarrow K_xCoTe_2 \tag{15}$$

$$3K_xCoTe_2 + (10 - 3x)K^+ + (10 - 3x)e^- \rightarrow 3Co + 2K_5Te_3 \tag{16}$$

$$K_5Te_3 + K^+ + e^- \rightarrow 3K_2Te \tag{17}$$

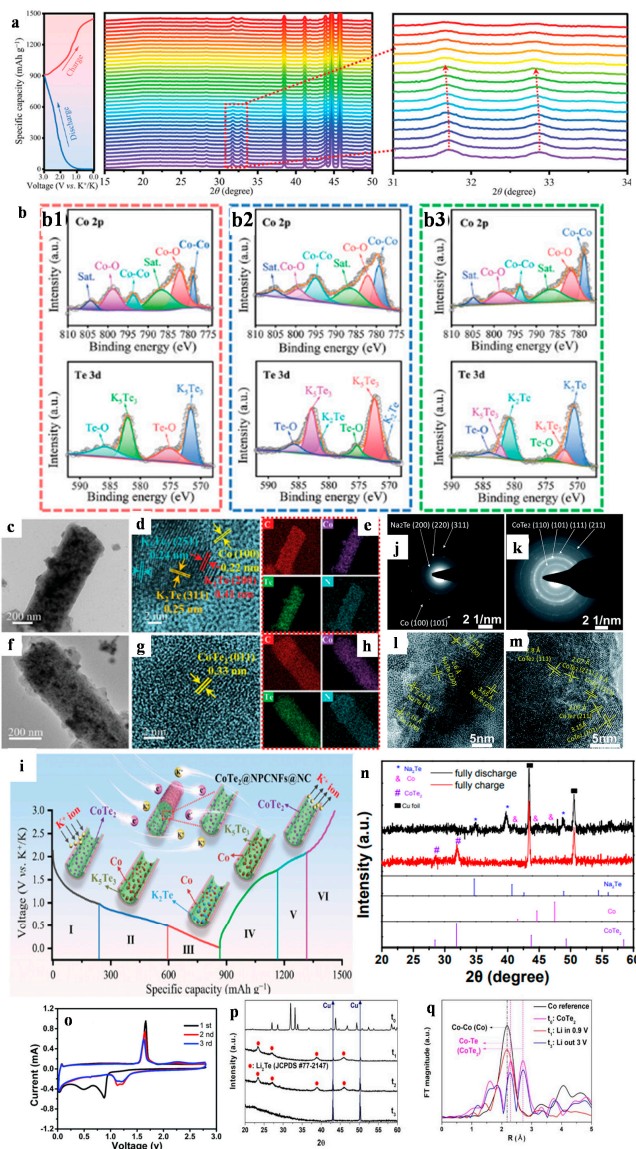

**Figure 4.** (**a**) In situ XRD patterns. (**b**) XPS of Co 2p and Te 3d at discharged state of (**b1**) 0.8 V, (**b2**) 0.4 V, and (**b3**) 0.1 V. (**c**,**f**) TEM, (**d**,**g**) HRTEM, and (**e**,**h**) EDS images at 0.1 V and 3 V, respectively. (**i**) Schematic diagram of the PIB storage mechanism of the $CoTe_2$@NPCNFs@NC; Ref. [41]. $CoTe_2$/G (**j**,**k**) SAED patterns, (**l**,**m**) HRTEM, and (**n**) ex situ XRD at 0.01 V and 2.8 V in the first cycle. (**o**) CV curve; Ref. [42]. (**p**) Ex situ XRD. (**q**) EXAFS after Li–embedded/Li–detached; Ref. [52].

Zhang et al. [42] clarified the sodium-ion storage mechanism of $CoTe_2$ by selected-area electron diffraction (SAED, Figure 4j–k), HRTEM (Figure 4l–m), and ex situ XRD (Figure 4n). At the end of discharge, the peaks of Co and $Na_2Te$ could be clearly observed from the ex situ XRD pattern. Moreover, the signals of Co and $Na_2Te$ could be detected in the SAED pattern. The crystal lattice of $Na_2Te$ and Co also appeared in HRTEM. After charging to 2.8 V, the $CoTe_2$ phase peak could be collected again. A clear $CoTe_2$ ring was observed in the SAED diagram. At the same time, Zhang analyzed the storage mechanism of Na ions from the CV curve (Figure 4o). A reduction peak formed at about 0.83 V, because $CoTe_2$ formed Co nanocrystals and $Na_2Te$ with Na ions. The oxidation peak at about 1.65 V can be ascribed to the reformation of $CoTe_2$. It can be summarized as follows:

$$CoTe_2 + 4Na^+ + 4e^- \leftrightarrow Co + 2Na_2Te \tag{18}$$

$$Co + 2Na_2Te \leftrightarrow CoTe_2 + 4Na^+ + 4e^- \tag{19}$$

Ganesan et al. [52] studied the deep material changes of $CoTe_2$ cycling in LIBs by ex situ XRD (Figure 4p) and extended X-ray absorption fine structure (EXAFS, Co-K-Edge, Figure 4q). In the XRD pattern, when discharging to 0.9 V, it can be seen that the peak of $CoTe_2$ changed to the peak of $Li_2Te$, and $CoTe_2$ also changed to Co in EXAFS, which means that the conversion reaction in the discharge process is that $CoTe_2$ changes to Co and $Li_2Te$. Finally, although no recovery of $CoTe_2$ was observed in the XRD pattern at a full charge of 3.0 V, the main $CoTe_2$ EXAFS peak reappeared, confirming the reversibility of the material. Based on the experimental results, the LIB conversion mechanism for $CoTe_2$ is proposed as follows:

$$CoTe_2 + xLi^+ + xe^- \leftrightarrow Co + Li_2Te \tag{20}$$

It is known that many works have fully studied the reaction mechanisms of cobalt selenide and cobalt telluride in LIBs, SIBs, and PIBs. Many in situ and ex situ analytical characterization methods have emerged. In situ XRD, XPS, and ex situ TEM, HRTEM, etc., can well demonstrate the mechanisms of $CoSe_x$ and $CoTe_x$. However, due to emerging anode design methods such as doping and heterojunction, heteroatoms and conductive carbon materials will bring various synergistic effects to cobalt selenide and cobalt telluride. At present, the mechanistic analysis of cobalt selenide and cobalt telluride anodes mainly focuses on the reaction between alkali ions and $CoSe_x$ or $CoTe_x$. In fact, the analysis of the synergistic mechanism between substances in the anode requires more in-depth research, and at the same time, innovative research methods need to be developed.

## 3. Electrode Design

As conversion-type anodes, $CoSe_x$ and $CoTe_x$ can bring enthralling high weight/volume capacity due to multiple electron transfer processes. However, due to the low conductivity of the converted anode materials and the large volume change in the process of intercalation/deintercalation, their practical application has been hindered. Therefore, effective strategies to solve these problems have been explored through carbon recombination [53], core–shell structures [54], defect engineering [55], etc.

Two approaches used to modify cobalt selenides and cobalt tellurides are the introduction of defects (e.g., point defects and heterogeneous interfaces) and structural design. Point defects like vacancies and dopings can interfere with surrounding atoms, resulting in lattice distortion, forming a good adjusted electronic structure with improved conductivity and diffusion coefficients [56]. The inorganic heterostructure of the anode builds an interfacial electric field as a facilitative transport channel, resulting in rapid Li/Na/K ion reaction kinetics. Considering the excellent prospects of interface engineering, Dong et al. [57] realized a "structural function motifs" design in a ZnO/ZnS anode. The ZnO/ZnS heterostructure regulates the state around the Fermi level, narrowing the band gap and, thus, improving the electronic conductivity. The introduction of defect engineering can create synergistic effects that alter the inherent properties of materials, unlike structural design

and composite strategies [58,59]. In addition, Co 3d electrons with the low spintronic structure t2g6eg1 are known to produce a Jahn–Teller effect in $CoSe_2$ [60]. Proper doping of metal ions can alleviate Jahn–Teller distortion to a certain extent [61,62] and promote a more stable electrode structure and better cycling stability. Defect engineering is widely used in $CoSe_2$ and has also been reported in recently emerging $CoTe_2$ materials.

In addition to defects, structural designs such as core–shell structures or $CoSe_x$ and $CoTe_x$ composite carbon materials [42] can take advantage of carbon's high conductivity and deformation resistance, not only reducing the risk of electrode crushing but also improving the rate performance. The latest reports of introducing defects or innovative structural designs to $CoSe_x$ and $CoTe_x$ are summarized below.

### 3.1. Introducing Defects

The modification methods of defect engineering include entry-point defects and surface defects. In recent studies, the introduction of point defects has been divided into metallic elements and non-metallic elements. Secondly, surface defects generally introduce heterogeneous structures. Recent advances in the combination of the above engineering with cobalt selenide and cobalt telluride are summarized below.

### 3.1.1. Doping with Metal Elements

Due to the large volume variation during circulation, the structural solidity and circulation function of metal selenides are weak, which limits the feasibility of their application. To overcome the above problems, some studies have prepared polyselenide metal nanostructures by adding inert elements [63]. The activity of the transition-metal-based materials can further be improved through coupling with other metals and non-metals [64]. The reason for the excellent doping performance of metal atoms is that the introduction of two or more metal atoms into a mixed-metal selenide results in the formation of more multicomponent selenides. Some studies have shown that this result can significantly improve the electrical conductivity of discharge products and further provide a buffer matrix to adapt to the expansion stress [65]. They benefit from strain relief and spectacular pseudocapacitance effects, thus reducing the energy barrier for ionic diffusion and showing considerable improvement in electrochemical performance.

For the voltage window of $CoSe_2$, previous studies have usually shortened it to 0.5~3.0 V to stabilize the reversible capacity of long cycles [66,67]. Ji et al. [50] doped a variety of metal ions such as $Ni^{2+}$ and $Cu^{2+}$ into nitrogen-doped carbon-shell-coated $CoSe_2$ particles to solve the problem of capacity loss caused by shortening the window. The doped product CoM–$Se_2$@NC, obtained by carbonizing a bimetallic zeolite imidazole skeleton (ZIFs) precursor, had more outstanding capacity than CoSe2@NC at 0.01~3.0 V. It was shown that the long cycle stability of $CoSe_2$ can be optimized effectively and the inefficient cycle caused by window adjustment can be eliminated under the synergistic action of the participation of another metal ion and the N-doped carbon shell.

Lu et al. [68] introduced Mo into a $CoSe_2$ self-supported nanosheet array (Mo-$CoSe_2$@NC) and investigated the performance of Mo in SIBs. In the XPS diagram (Figure 5a–b), the 3d spectrum of Mo consisted of peaks at 232.3 eV and 235.4 eV, associated with the characteristic spin-orbital bistates of Mo $3d^{5/2}$ and $3d^{3/2}$, respectively, which implies that $Mo^{6+}$ ions enter the $CoSe_2$ lattice [69]. The embedded Mo defect in the $CoSe_2$ lattice not only produced more redox sites, but also induced the generation of mixed phases of o-$CoSe_2$ and c-$CoSe_2$, as evidenced by the coexistence of two hybrid phases of $CoSe_2$ in the SAED diagram (Figure 5d).

Wang [70] formed FeCo-Se in situ on a carbon cloth, and this multilayer array was able to provide sufficient active surfaces and shuttle channels for sodium ion exchange, thereby achieving excellent electrochemical kinetics. XRD (Figure 5g) proved that after the Co-Se was doped with Fe, impurities such as cobalt oxide, iron oxide, iron selenide, and FeCo alloy were not produced, implying that Fe ions only enter the Co-Se lattice. Notably, when more than 50% iron salt was added to the Co-Se material, the morphology of Fe-Se changed

from a layered nanosheet array into pyramidal particles (Figure 5e–f), with a decline in electrochemical performance.

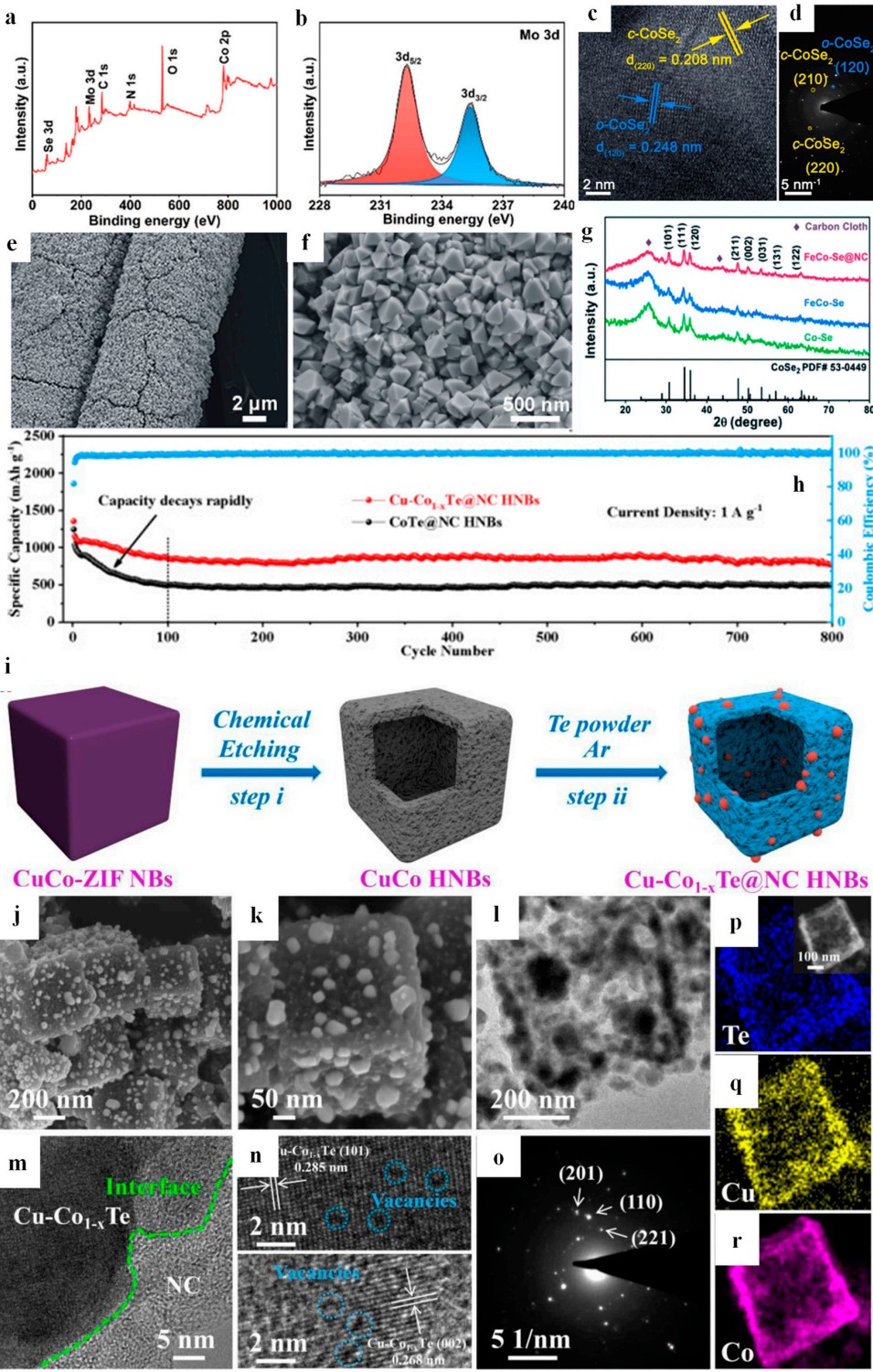

**Figure 5.** XPS of Mo–CoSe$_2$@NC with (**a**) survey and (**b**) Mo 3d. (**c**) HRTEM and (**d**) SAED; Ref. [68]. (**e**,**f**) Typical low– and high–magnification SEM (scanning electron microscope) images and (**g**) FeCo–Se XRD; Ref. [70]. (**h**) Li–ion batteries' performance. (**i**) Schematic representation of the synthesis of Cu–Co$_{1-x}$Te@NC HNBs with (**j**–**l**) SEM and TEM images, (**m**,**n**) HRTEM images, and (**o**) SAED pattern and (**p**–**r**) elemental mappings, Ref. [71].

### 3.1.2. Doping with Non-Metallic Elements

For cobalt telluride, Hu et al. [71] prepared Cu-doped cobalt telluride hollow carbon nanocells (Cu-Co$_{1-x}$Te@NC HNBs, Figure 5j–l) by chemically etching CuCo-ZIF nanocells and tellurizing them (Figure 5i), The copper–cobalt tellurium was homogeneously embedded in the nanobox (Figure 5p–r), and Co vacancies were constructed by copper induction (Figure 5n). At the same time, the interaction in the heterogeneous interface between nitrogen-doped carbon and cobalt telluride (Figure 5m) triggered the transfer of the p-band to interface charge transfer, so that the composite exhibited faster ion and electron diffusion kinetics than HNB electrodes, contributing to improved storage performance of lithium-ion batteries (Figure 5h).

Doping with non-metallic elements, especially chalcogenide elements, can not only produce polysulfides with alkali ions to improve the storage capacity, but also induce vacancy formation like metal elements to form a built-in electric field. Yu [34] reported an octahedral CoSe$_2$ sawtooth chain array threaded with N-doped carbon nanotubes (NCNTs) as a high-performance PIB anode. Highly conductive NCNTs form a flexible conductive network that greatly accelerates electron transfer and improves rate performance. Nitrogen-doped NCNTs also provide additional capacity to CS-NCNTs, constituting a versatile composite carbon material.

Wang [33] prepared Co$_{0.85}$Se$_{1-x}$S$_x$ nanoparticles by using S, which is of the same family as Se, as a point doping tool. The sample was formalized in a metal–organic framework, carbonized in graphene and, finally, cured in situ. Density functional theory (DFT) proved that S doping enhanced the adsorption energy of Co to alkali metal ions (Figure 6a–h). The hollow polyhedral skeleton of the double-carbon shell (Co$_{0.85}$Se$_{1-x}$S$_x$ @C/G) with a stable structure and S vacancies reduced the volume change, increased the number of electrochemical active sites, and improved the alkali-ion storage performance.

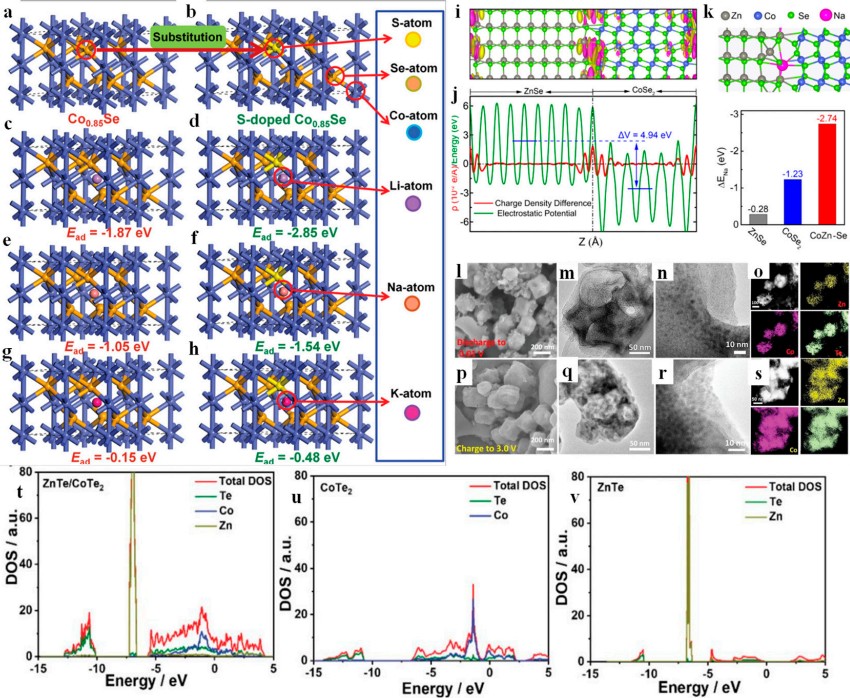

**Figure 6.** (**a–h**) DFT in crystal structures and alkali–ion adsorption sites for (**a,c,e,g**) Co$_{0.85}$Se and (**b,d,f,h**) S–doped Co$_{0.85}$Se; Ref. [33]. (**i**) The differential charge density between CoSe$_2$ and ZnSe in the phase boundary is calculated. (**j**) Averaged electrostatic potential. (**k**) The Na atom adsorption energy of ZnSe, CoSe$_2$, and CoZn–Se; Ref. [72]. (**l–o**) SEM, TEM, HRTEM, and EDS images, respectively, for a ZnTe/CoTe$_2$@NC anode at 0.01 V and (**p–s**) 3.0 V. (**t–v**) Total density of states (TDOS) for CoTe$_2$, ZnTe, and ZnTe/CoTe$_2$, respectively; Ref. [40].

Moreover, given the high conversion capacity of phosphorus, Ye et al. [73] reported in another work that P was successfully doped into $CoSe_2$. It was found through P 2p spectroscopy that additional Co-P and P-Se bonds could be generated by P doping, which could enhance the structural stability of $CoSe_2$. Xu [38] also used P to induce Te vacancies in o-$CoTe_2$ nanowires. Compared with h-$CoTe_2$, the Co 2p and Te 3d peaks of o-$CoTe_2$ showed significantly higher binding energy on the XPS images after P doping. EPR showed that the characteristic Te vacancy signal of o-P-$CoTe_2$/MXene was stronger and wider, indicating that the vacancy content increased significantly. Brunauer–Emmett–Teller (BET) theory showed that P doping increased the specific surface area. These results show that Te vacancy defects and P doping not only provide the active sites but also jointly enhance the structural stability of $CoTe_2$.

### 3.1.3. Heterojunctions

Heterostructure refers to the geometric structure and connecting interface formed by the combination of two or more materials through physical or chemical methods [74,75]. In the heterojunction, electrons will transition from a high Fermi level to a low Fermi level, resulting in potential difference due to interfacial polarization at the heterojunction [76] and the internal electric field. The existence of the internal electric field effect is helpful to accelerate the diffusion of metal ions and increase the adsorption energy of ions [77,78]. In addition, lattice distortion and charge redistribution taking place at heterogeneous boundaries can not only improve charge-transfer efficiency and conductivity, but also provide additional active sites for reversible redox reactions [79,80]. Therefore, due to the active role of the heterogeneous interface, heterogeneous anodes can obtain a stable nanostructure and excellent storage performance [81]. Therefore, in view of problems such as the low conductivity and large volume expansion of chalcogen-based metal materials, it is considered to be an effective strategy to construct high-quality heterogeneous structural interfaces to improve their storage performance [82].

Previously, Fang [72] reported a bimetallic selenide heterogeneous structure ($CoSe_2$/ZnSe@C) for SIBs. The phase-boundary charge redistribution of Co/Zn was studied by theoretical calculation. The $Na^+$ adsorption energy showed that the ZnSe side had a phase boundary with high electron density, which was more conducive to the adsorption of $Na^+$ ions, accelerating the ion movement and charge conduction, and achieving high reaction kinetics and high electrical conductivity (Figure 6i–k). In addition, the multistep conversion reaction in the Co/Zn heterostructure can effectively relieve the stress of $Na^+$ insertion. $CoSe_2$/ZnSe@C also showed good OER (oxygen evolution reaction) activity, providing additional evidence that the Co/Zn heterogeneous interface accelerated the redox reaction.

Xiao [83] recently also reported a similar application of heteroselenides in SIBs. He developed ZnSe/$CoSe_2$-CN heterostructures with Se vacancies via controllable in situ selenization. HETEM images showed a clear heterogeneous interface between ZnSe and $CoSe_2$, which can create electric fields to promote the movement of ions and electrons. Electron paramagnetic resonance (EPR) spectra showed an obvious signal caused by the selenium vacancy trapping electrons at g = 2.003, confirming the presence of the Se vacancy in the sample. The more induced active sites and the enhanced electronic conductivity introduced by the Se vacancy can improve the electrochemical properties of the materials.

Analogously, Zhang et al. [40] reported a template-assisted strategy to achieve in situ formation of a bimetallic tellurium heterostructure (ZnTe/$CoTe_2$@NC) on N-doped carbon shells. The results showed that both the electron-rich Te sites and the internal electric fields generated by electron transfer from ZnTe to $CoTe_2$ in Te heterojunctions provided rich cation-adsorption sites and promoted interfacial electron transport during the potassic/de-potassic process. Notably, as Zn led to ZIF-8 crystal formation, with the introduction of $Zn^{2+}$ species into Co-ZIF-67, $Zn^{2+}$ ions acted like an obstacle to Co-ZIF-67 crystals' nucleation and growth. Concretely, Zn delays the Co-ZIF-67 nucleation dynamics, thus resulting in relatively small doping particle sizes. The growth of ZIF-67 dominated by Co leads to the production of larger crystals in the reaction–diffusion

process [84,85]. The finer $ZnTe/CoTe_2$ nanoparticles prepared by the heterojunction method showed higher structural stability. Ex situ SEM, HRTEM, and EDS were performed on the cycled $ZnTe/CoTe_2$@NC (Figure 6l–s). The ZnTe and $CoTe_2$ nanoparticles kept their original appearance, with no obvious particle aggregation. In addition, the TDOS results presented in Figure 6t–v highlight the discontinuous band gap of Fermi levels and confirm the semiconductor properties. In contrast, $ZnTe/CoTe_2$ has an obvious hybridization zone in the conduction band, resulting in significantly higher TDOS of $ZnTe/CoTe_2$ than that of both ZnTe and $CoTe_2$. This indicates that the electronic conductivity of $ZnTe/CoTe_2$ is significantly improved.

In addition to zinc–cobalt heterojunctions, He et al. [86] chose Ni to produce coral-like $Ni_xCo_{1-x}Se_2$ for the first time using a layered structure. The introduction of Ni formed a hierarchical structure to prevent structural fragmentation, accelerated the electron transmission, and shortened the $Na^+$ diffusion path. Zhu et al. [87] used the carbonization and subsequent selenization process design to encapsulate $Ni_3Se_4$@$CoSe_2$@C/CNTs in a core–shell three-dimensional interpenetrating two-carbon frame. $Ni_3Se_4$@$CoSe_2$ was evenly dispersed into a three-dimensional carbon skeleton structure/carbon nanotube network, greatly enhancing the conductivity and further achieving fast Na+ diffusion.

*3.2. Structural Design*

Nanomaterials, especially spherical and small nanocrystals, have their isotopic physical properties, large active surface, and high packing density [88–90]. More active sites can be exposed by delicate structures (e.g., sea urchin $CoSe_2$ [88], CoSe microspheres [91]). By using carbon coatings, $CoSe_x$ and $CoTe_x$ can be well maintained in nanoform and improved in terms of cycling stability by using constraint effects (such as CoSe-C@C [92] and o/h-$CoTe_2$ [93]). Combination with structural carbon can produce interfacial interactions, which can effectively improve diffusion kinetics, electrical conductivity, and structural compatibility, thus achieving excellent electrochemical performance [10,94]. For example, three-dimensional conductive carbon network structures with large channels can provide diffusion paths, improve the migration rate of ions in the battery material, and reduce the limitation of poor conductivity of $CoSe_x$ and $CoTe_x$ [66]. Metal–organic frameworks (MOFs) possess exceptionally high surface area and porosity, allowing for the adsorption and storage of gases, liquids, and even small molecules. This property makes them promising for applications such as energy storage, separation, and catalysis. Moreover, by optimizing the composite structure, the electrochemical properties of the composites can be further improved, and the volume strain of the active nanomaterials in the carbon matrix can be reduced. Therefore, carbon materials such as reduced graphene oxide (rGO) and MXenes are not only electrically conductive but also have good ductility, which can greatly enhance the structural compatibility of $CoSe_x$ and $CoTe_x$ anode materials.

However, MOFs, MXenes, and other carbon-based materials have their limitations. For example, MOFs are prone to acid–base interference and structural collapse. MXene etching processes can be complex and hazardous. Additionally, low energy density is a common drawback of these materials. Therefore, recently, researchers have been consolidating the carbon-based structure of transition-metal-based MOFs through annealing and synthesizing high-performance ion battery anodes using vapor-phase methods [95]. High-capacity ion battery anode materials can be obtained through green etching of MXenes using only grinding and annealing via molten salt methods [96]. These structural design approaches are frequently employed in the design of cobalt selenide and cobalt telluride anodes. The latest progress in combining the abovementioned methods with cobalt selenide and cobalt telluride for alkali-metal-ion battery cathodes is summarized below.

3.2.1. Freestanding Electrodes

Common electrodes are made of an active material composited with a binder and conductive carbon on the current collector, and the presence of the binder and collector fluid especially affects the energy density. Therefore, the development of freestanding

electrode structures to increase the proportion of active materials and eliminate the negative effects of the binder is a promising direction [97].

Electrospinning is used to produce individual electrode materials, allowing the active material to be embedded in a network of carbon fibers. The carbonized material can be cut so as to be used as an electrode, avoiding mechanical losses caused by mixing adhesives with conductive materials and coatings [98,99]. Self-supported electrodes can alleviate electrode pulverization and construct interwoven electron/ion transport networks, thus improving electrochemical energy storage performance.

Zhan et al. [47] innovatively prepared a self-supported cobalt telluride electrode for SIBs. They synthesized $CoTe_2$@NMCNFs via electrospinning and subsequent in situ tellurization (Figure 7a–f). Material characterizations revealed NMCNFs with N-doped active sites and porous carbon networks with high specific surface area. These positive effects promote electrolyte penetration and mitigate the aggregation effect in the $CoTe_2$ nanoparticles by reducing the volume changes during the cycle (Figure 7g–j). The long cycle of $CoTe_2$@NMCNFs shows a high reversible $Na^+$ storage capacity, long-term cycle stability, and high rate capacity. In general, the overall conductivity of the electrode is greatly improved, and $Na^+$ diffusion is promoted, thanks to the strong three-dimensional crosslinked composite fiber and the absence of voluminous inactive components.

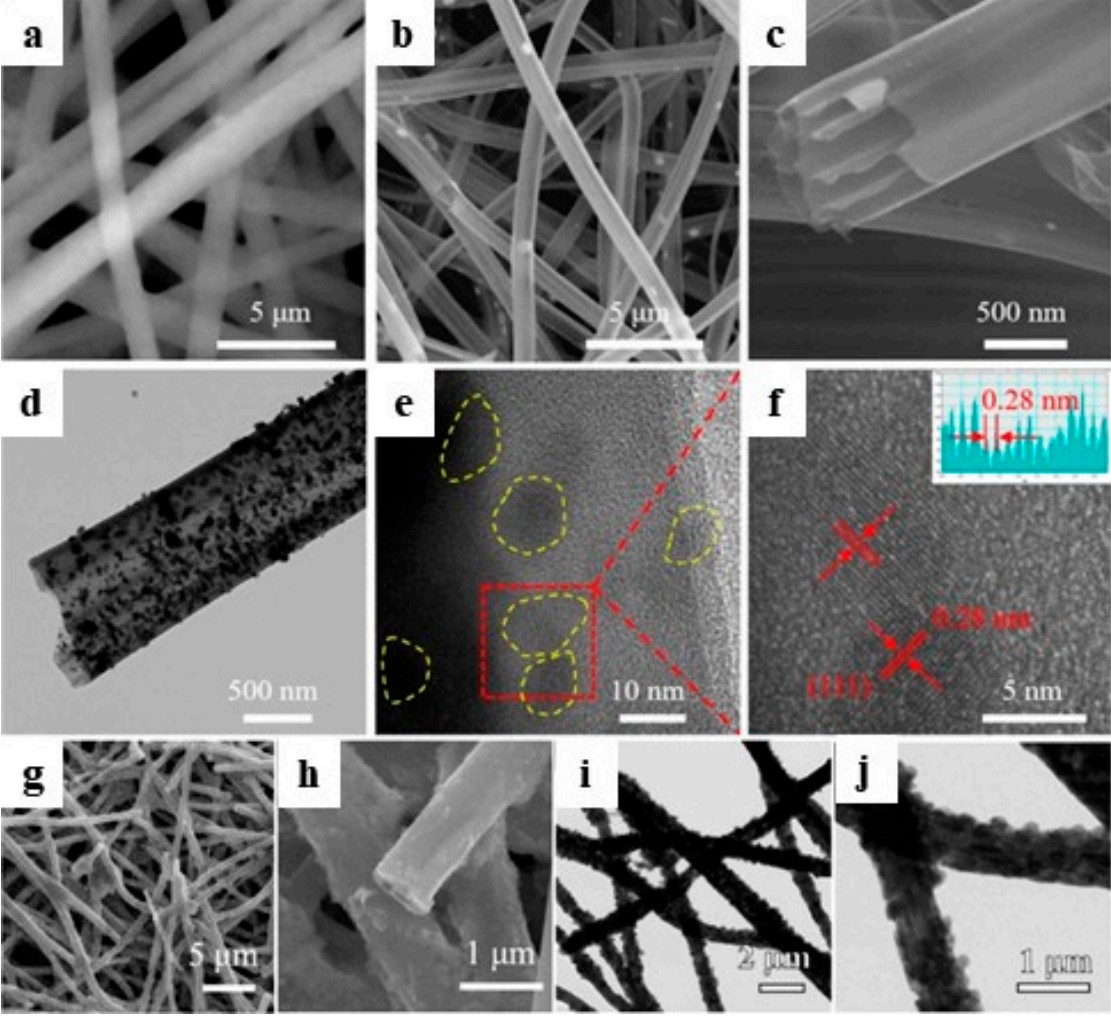

**Figure 7.** (**a–c**) SEM and (**d–f**) TEM images of PAN/PS/Co(ac)$_2$·4H$_2$O. (**g,h**) SEM and (**i,j**) TEM images of $CoTe_2$@NMCNFs after 300 cycles at 200 mA g$^{-1}$. Ref. [47].

Meanwhile, Park et al. [100] prepared cobalt selenite (p-CoSeO$_3$-CNF) embedded in carbon composite nanofibers by electrospinning. Among them, the heat-treated nanofibers containing cobalt and selenium formed CoSe$_2$ nanocluster intermediates. Then, through the final oxidation step, CoSe$_2$ was converted to amorphous CoSeO$_3$. Part of the carbon fiber was changed into CO$_2$ gas during the oxidation process, leaving pores and forming mesopores in the nanofibers. The carbon and pores limited the volume change of CoSeO$_3$ during repeated sodification/desodification processes, allowing the electrolyte to permeate easily and reducing the diffusion path of sodium ions. The large network carbon structure accelerated electron transport, and the high performance of p-CoSeO$_3$-CNF was demonstrated. Electrospinning has also been reported for SIB and LIB anode materials other than CoSe$_2$ [32,101].

### 3.2.2. Carbon-Cloth-Based Electrodes

Commercially available carbon cloth (CC) material, consisting of hundreds of fibers woven tightly together, ensures that the fibers are interconnected and can be rolled into different shapes without compromising their integrity [102]. The stable cycling performance of CC-based electrodes makes them a competitive option for industrial applications of metal-ion batteries [103].

According to the different raw materials, carbon fiber is mainly divided into three substrates: asphalt-based carbon fiber, PAN-based carbon fiber, and viscose-based carbon fiber. PAN-based carbon fiber is rich in raw materials and superior to the other two types. Therefore, PAN-based carbon fiber is the most widely used, accounting for more than 90% [104]. The difference between the growth of active substances in PAN-based carbon cloth and the embedding of active substances in electrospinning is that carbon cloth generally penetrates active substances into existing carbon substrates via infiltration methods, hydrothermal methods, and other methods. However, electrospinning uses PAN, solvent, and metal salt to dissolve directly into the active material. Both have their advantages, such as the toughness of carbon cloth and the advantage of electrospinning for in situ doping treatments, which can enhance the synergistic effects of carbon nanowires and active substances [105].

To date, carbon cloth combined with active substances has been used in fuel cells [106], catalysis [107], lithium–sulfur batteries [108,109], supercapacitors [102], etc. There have also been some studies on carbon cloth combined with metallic chalcogenide compounds as negative electrodes for LIBs [110]. For cobalt selenide, Wang et al. [70] grew FeCoSe on carbon cloth, and Lu et al. [48] synthesized Mo-doped CoSe$_2$ nanosheets, proving that the carbon cloth and cobalt selenide have a good synergistic effect and are more conducive to the formation of bimetallic synergies (Figure 8). However, cobalt telluride combined with carbon cloth as a negative electrode for LIBs, SIBs, and PIBs has rarely been reported.

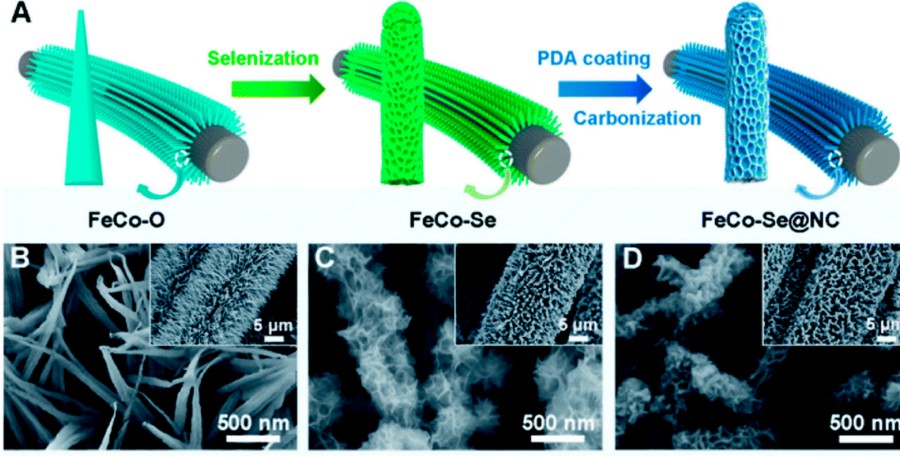

**Figure 8.** (**A**) Synthesis of FeCo–Se@NC and (**B**–**D**) SEM images of the material in each step; Ref. [68].

### 3.2.3. MOF-Based Electrodes

MOFs with a high specific surface area and adjustable pore structure are widely used in the fields of calcification and energy conversion due to their stable structure, large specific surface area, and diverse functions [95,111,112]. Under appropriate conditions, MOFs can be used as precursors and/or templates to derive metal chalcogenides with regular topography and excellent electrochemical properties (Figure 9) [113–115]. Notably, MOFs have proven to be ideal templates for the preparation of transition metal selenides and tellurides. In recent reports, it has been shown that transition metal selenide and telluride nanocrystals can be embedded 1–3D porous carbon substrates to form efficient conductive networks [116]. In addition, the relatively high surface area and large pores of the high-dimensional conductive substrate enhance the wettability of the electrolyte to the material and promote ion diffusion, thereby improving the storage performance [117]. Therefore, MOFs as a template for the preparation of nanoscale electroactive materials and porous carbon matrix hybrid structures have broad application prospects.

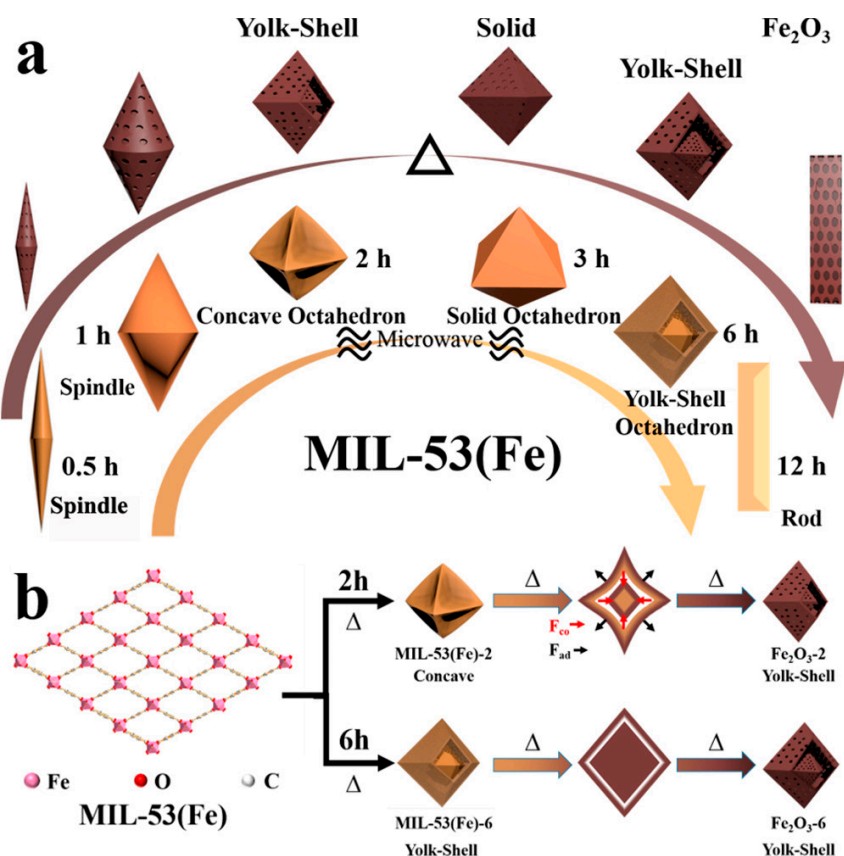

**Figure 9.** Transition metals have various valence states and can form various complex structures, such as Fe–based MIL–53 with yolk–shell octahedron morphologies: (**a**) process of MIL–53(Fe) growth and (**b**) the formation process of $Fe_2O_3$–2 and $Fe_2O_3$–6; Ref. [115].

More recently, Xiao [39] prepared hollow $In_2Se_3$/$CoSe_2$ nanorods by growing Co-ZIF-67 on the surface of In-MIL-68 and via in situ selenization. During selenization, selenium atoms penetrate from the surface of the nanorod to the inside, while indium atoms diffuse outward in opposite directions and at different speeds, creating the Hilken–Dahl effect and forming hollow structures [118]. At the same time, the mass ratio of ZIF-67 to MIL-68 grown on the surface was studied by changing the concentration of cobalt nitrate. The results showed that the ZIF-67 nanoparticles became smaller and could be uniformly deposited on the surface of MIL-68 with the decrease in the concentration of cobalt nitrate in solution (Figure 10).

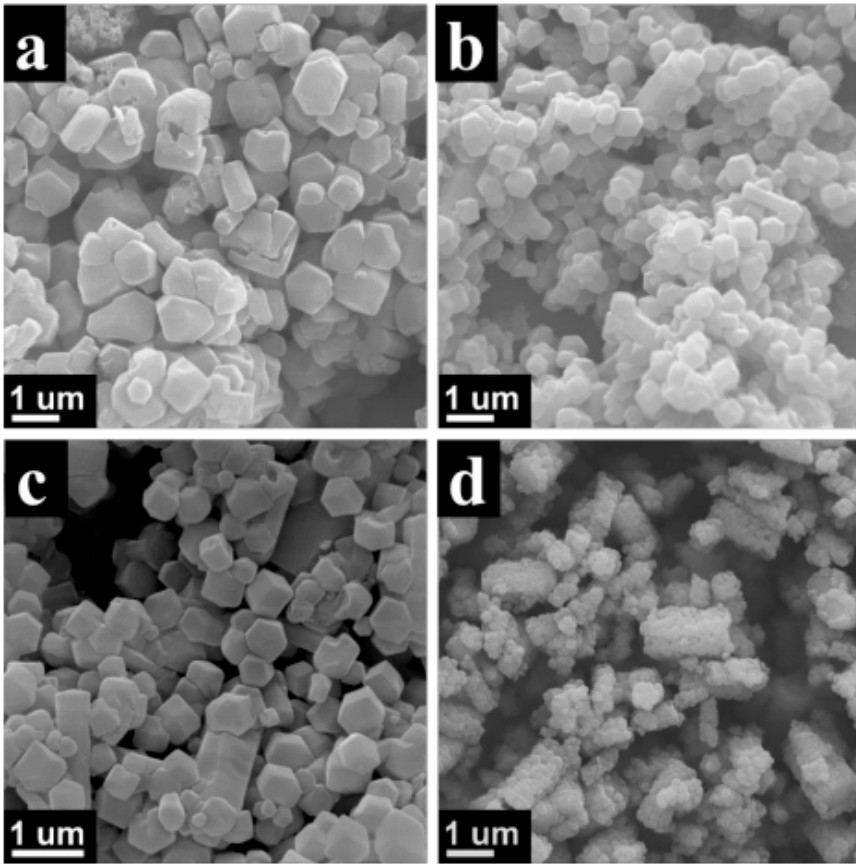

**Figure 10.** SEM images of In-MOF: Co $(NO_3)_2$ = (**a**) 1:8, (**b**) 1:6, (**c**) 1:4, and (**d**) 1:3; Ref. [39].

Wang et al. [119] prepared cobalt selenide (CoSe$_2$-CNS) anchored on carbon nanosheets by selenizing cobalt embedded in 1D-MOF carbon nanosheets (Co-CNS). CoSe$_2$ grown in situ on carbon nanosheets showed close contact between CoSe$_2$ and carbon. At the same time, CoSe$_2$-CNS also has typical two-dimensional structural characteristics, which are conducive to electrolyte penetration and ion/electron transport. In addition, the carbon matrix can alleviate the volume expansion of CoSe$_2$ during the desodification process. As the result of these advantages, CoSe$_2$-CNS provides a stable cycle capacity of 468 mA h g$^{-1}$, together with a high rate capability of 352 mA h g$^{-1}$ at 10,000 mA g$^{-1}$.

3.2.4. MXene-Based Electrodes

MXenes, which include transition metal carbides or carbon nitride, have proven to be novel materials with great potential for energy storage [120,121]. MXenes have high electrical conductivity, large interlayer spacing, and excellent mechanical properties, enabling them to act as a fluid collector and buffer deformation at the same time [122]. Their electrical and electronic properties can be customized through modifications to the MXenes' functional groups, solid solution structures, or stoichiometry to meet the needs of different scenarios [123]. As a result, titanium carbide has demonstrated outstanding catalytic activity and selectivity as a co-catalyst in various catalytic reactions, finding wide applications in energy conversion, environmental protection, and organic synthesis [124].

Xu [38] used P-induced o-CoTe$_2$ nanowires to anchor MXene nanosheets (o-P-CoTe$_2$/ MXene). The elastic MXene formed a tight interfacial interaction with P-CoTe$_2$ (Figure 11). DFT calculations further showed that the new superstructure had higher electronic conductivity, enhanced the adsorption capacity of K$^+$ ions, and reduced the energy barrier to ion diffusion. Meanwhile, the entire battery was cycled through with negligible capacity loss even after bending. This proves that the combination with MXenes can greatly improve the stability of the material. Hong [91] found a strong Co-O-Ti covalent interaction in

CoSe$_2$@MXene. It assembled the inner hollow CoSe$_2$ microsphere and the outer MXene via electrostatic self-assembly. This covalent bond facilitates electron/ion transport and enhances the structural durability of the CoSe$_2$@MXene hybrid, thus improving the rate performance and cyclic stability. It has been reported that the polar surface of MXenes can inhibit the polysulfides' shuttle effect [125,126]. Wang [127] was therefore inspired to grow CoSe$_2$ nanorods in situ on the surface of an MXene via a simple one-step hydrothermal method. Importantly, the stability of the CoSe$_2$/MXene cycle in this study was better than that of the pure CoSe$_2$ cycle, demonstrating that Ti$_3$C$_2$T$_x$ can inhibit the undervoltage failure caused by Na$_x$Se instability.

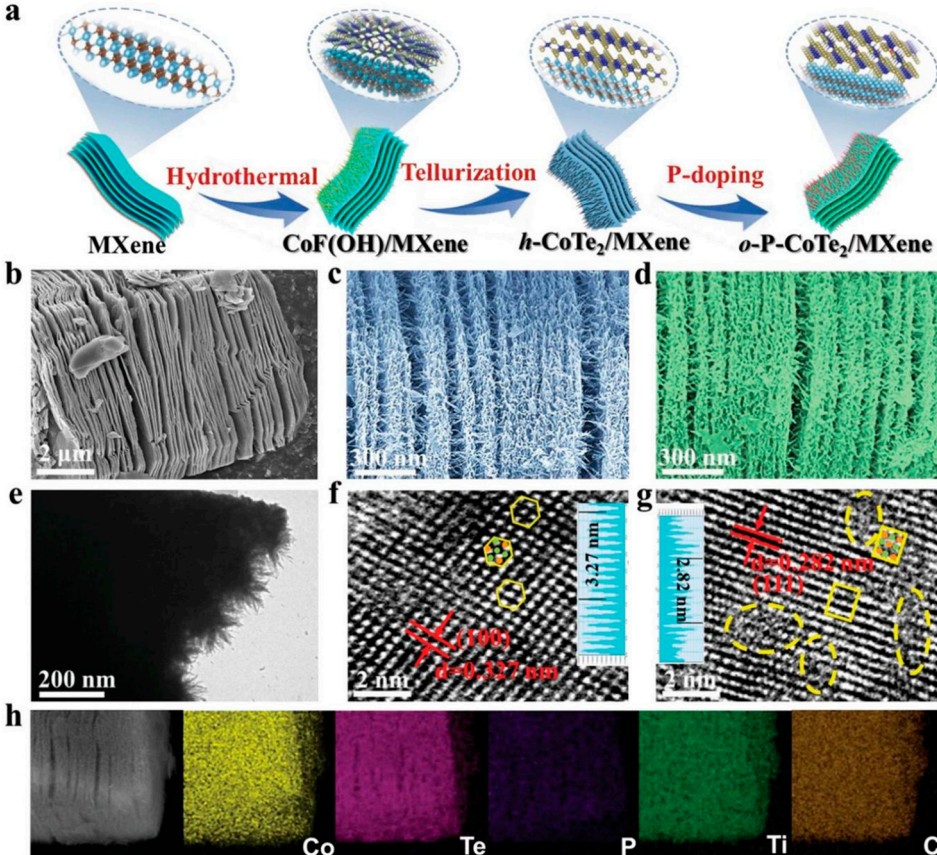

**Figure 11.** (**a**) Preparation process. SEM images of (**b**) MXene, (**c**) h–CoTe$_2$/MXene, and (**d**) o–P–CoTe$_2$/MXene. (**e**,**g**) TEM and HRTEM images of o–P–CoTe$_2$/MXene; (**f**) HRTEM of h–CoTe$_2$/MXene. (**h**) EDS spectrum of o–P–CoTe$_2$/MXene; Ref. [38].

### 3.3. Summary of Electrode Design

Cobalt selenide, as a high-capacity conversion anode, has been widely studied in alkali-metal-ion batteries. The new electrode design methods, such as doping with composite conductive materials (e.g., MXenes), can show excellent high capacity and high stability. However, the wide application and innovation of cobalt selenide in potassium-ion batteries still need to be improved. Cobalt telluride, as a compound of the same series, is still in its infancy in the design of advanced anode materials. Some novel designs of cobalt telluride show excellent electrochemical performance, which indicates the great potential of cobalt telluride. However, innovative designs of cobalt selenide are still to be developed; for example, co-doping with multiple non-metallic elements, carbon cloth growth, and electrospinning in situ growth methods have not been tried. At the same time, the electrochemical reaction mechanism of various synergistic effects is still unclear, and products with ultra-long periods of stability are still to be developed.

## 4. Electrode Synthesis

In addition to the elemental composition of the material, the morphology, crystal phase, and structure of the material also have a great influence on its properties. The microstructure of materials, such as their size and morphology, will affect their electron energy density, as well as influencing their optical and electrical properties [90,107,108,128]. Therefore, choosing appropriate preparation methods to synthesize materials with special structures is the key to improving the properties of the materials. Conventional preparation methods of cobalt-based anode materials include, but are not limited to, hydrothermal/solvothermal methods [32,101,113,116], heat treatment methods [52,93,129], sol–gel methods [130], vapor deposition methods [34], and electrospinning [100]. At the same time, the usual method for preparing selenides and tellurides is the gas-phase method [34].

Hydrothermal reaction is a typical synthesis route. Under certain temperature and high-pressure conditions, cobalt-based materials with high phase purity and crystallinity can be synthesized. For example, Lei et al. [131] synthesized uniform $CoTe_2$ nanorods by the hydrothermal method, without any surfactant or mineralizer for 20 h, using ascorbic acid as a reducing agent. The study found that simply adjusting the concentration of sodium hydroxide can produce different forms, including short rod-shaped bodies with many spines and flower-like hierarchies (Figure 12a–f). Hydrothermal methods are widely used in the synthesis of various materials, due to their advantages of fewer experimental steps, changeable morphology, and simple operation. However, hydrothermal methods also have some shortcomings, such as too many uncontrollable factors, easily leading to an uneven appearance, and a generally long hydrothermal reaction time. As a high-temperature treatment, the vapor-phase method can be used alone or in combination with a template method for intermediate processes such as selenides and tellurides. Cobalt-based electrode materials with an abundant pore structure and high electrochemical performance can be obtained by heat treatment. Heat treatment is almost a necessary experimental method for MOF-series materials, but this method has high energy consumption, and for MOF materials the sintering temperature is too high, which can easily lead to the collapse of the structure. The above methods are conducive to the formation of high-functional structures for $CoSe_x$/$CoTe_x$, but high-temperature pyrolysis is always necessary in the process of in situ carbonization. To achieve energy savings and high yields, it is urgent to explore more effective modification or combination methods such as magnetron sputtering [132], microwave [133], and molten salt etching [96,134] methods. The next section summarizes the current $CoSe_x$/$CoTe_x$ anode synthesis methods for LIBs, SIBs, and PIBs.

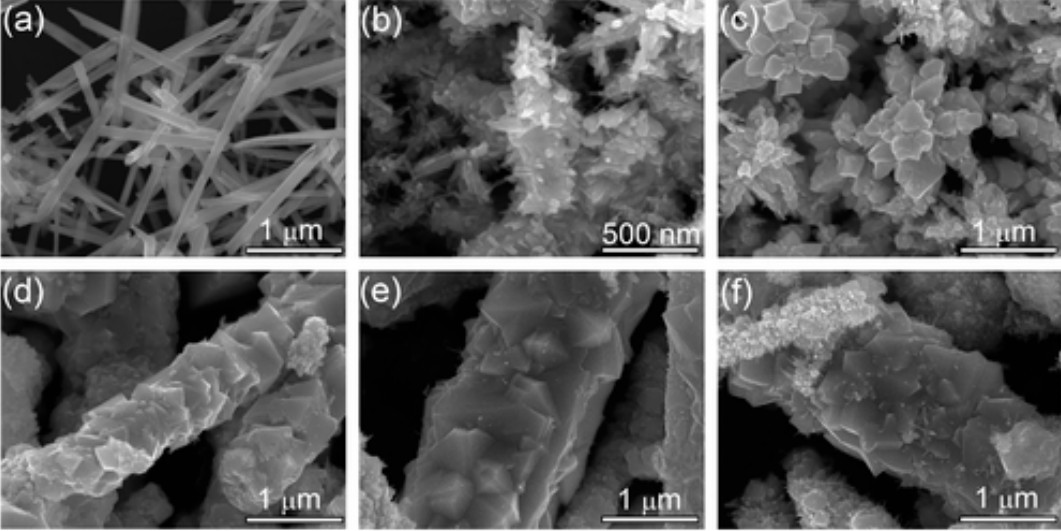

**Figure 12.** (**a**–**f**) SEM for $CoTe_2$ synthesized with different NaOH concentrations: 0.0 M, 0.2 M, 0.5 M, 1.2 M, 1.5 M, and 2.0 M. Ref. [131].

## 5. Performance Summary

This section focuses on the latest advances in the synthesis of various cobalt-based selenides and tellurides for LIBs, SIBs, and PIBs, and it summarizes the key properties and synthesis methods of various heterostructured anodes, as shown in the following Tables 1 and 2.

**Table 1.** Review of the properties of $CoSe_x$ electrode materials.

| Sample Name | Synthesis Method | Structure | Cycle Performance (mAh g$^{-1}$, A g$^{-1}$) | Rate Performance (mAh g$^{-1}$, A g$^{-1}$) | Initial Coulombic Efficiency (%, A g$^{-1}$) | Battery Type | Ref./Year |
|---|---|---|---|---|---|---|---|
| $CoSe_2$/carbon nanoboxes | Annealing | Nanoboxes | 860/0.2/100 th 660/1/100 th | 686/2 | 78.3%/0.2 | LIB | [135]/2016 |
| CoSe@NCNTs | Annealing | Nanowires | 326/0.5/100 th | 278/3 | 95%/0.1 | PIB | [18]/2021 |
| $CoSe_2$-CNS | Annealing | Nanosheets | ~250/10/2000 th | 352/10 | | SIB | [103]/2019 |
| $CoSe_2$@NC/ CNTs | Annealing | Nanowires | 480/1/200 th 369/10/800 th | 360/10 | | SIB | [136]/2022 |
| Mo– $CoSe_2$@NC | Electrodeposition + selenization | Self-supporting nanosheet | 672/0.5/200 th | 313/5 | 47.9%/0.5 | SIB | [68]/2022 |
| $In_2Se_3$/$CoSe_2$– 450 | Annealing | MIL-68@ZIF-67 core–shell | 445.0/0.5/200 th 297.5/5/2000 th 205.5/10/2000 th | 371.6/20 | 61.2%/0.1 | SIB | [39]/2021 |
| TNC–$CoSe_2$ | Precipitation + sol–gel + annealing | Microcubes | 511.2/0.2/200 th | 464/6.4 | 88.26%/0.2 | SIB | [130]/2021 |
| ZnSe/$CoSe_2$– CN | Annealing | Spherical particles | 547.1/0.5/300 th 422.6/3/300 th | 362.1/20 | 97.7%/0.1 | SIB | [83]/2022 |
| $CoSe_2$@MXene | Annealing | Porous hollow shell encapsulated by MXene nanosheets | 910/0.2/100 th 1279/1/1000 th | 465/5 | 71.9%/0.2 | LIB | [91]/2020 |
| $Ni_{0.47}Co_{0.53}Se_2$ | Solvothermal | Coral-like | 321/2/2000 th | 277/15 | 98.5%/0.1 | SIB | [86]/2019 |
| p–$CoSeO_3$– CNF | Electrospun + annealing | Composite porous nanofibers | 288/0.2/200 th | 207/5 | 56.7%/0.5 | SIB | [100]/2021 |
| $CoSe_2$/CoSe | Annealing | Hierarchical hollow raspberry-like superstructure | 1361/1/1000 th 579/2/2000 th 315/5/1000 th | 406/5 | | LIB | [54]/2019 |
| $Ni_{0.33}Co_{0.67}Se_2$ | Solvothermal selenization | Hierarchical mesoporous nanospheres | 301.9/1/1300 th | 314.5/8 | 85.4%/0.5 | SIB | [137]/2023 |
| $CoSe_2$/ZnSe | Annealing | Bimetallic heterostructure | ~250/8/4000 th ~200/10/4000 th | 263/10 | 72.3%/0.1 | SIB | [72]/2019 |
| CoSSe@C/G | Solvothermal + annealing | Double-carbon shells | 636.2/2/1400 th 353/1/200 th 208.1/0.2/100 th | 254.5/10 266.6/10 195.7/2 | 68.09%/0.1 71.2%/0.1 48.3%/0.1 | LIB SIB PIB | [33]/2020 |
| NCNF@CS | Chemical vapor deposition + solvothermal | Octahedral threaded N-doped carbon nanotubes | 253/0.2/100 th 173/2/600 th | 196/2 | 69.3%/0.2 | PIB | [34]/2018 |
| 3DG/ $CoSe_2$@CNWs | Hydrogel | Multidimensional porous nanoarchitecture | 543/0.1/100 302/2/500 th | ~320/2 | 63.5%/0.1 | SIB | [30]/2022 |

**Table 1.** *Cont.*

| Sample Name | Synthesis Method | Structure | Cycle Performance (mAh g$^{-1}$, A g$^{-1}$) | Rate Performance (mAh g$^{-1}$, A g$^{-1}$) | Initial Coulombic Efficiency (%, A g$^{-1}$) | Battery Type | Ref./Year |
|---|---|---|---|---|---|---|---|
| CoSe$_2$@NC/rGO-5 | Two-step co-precipitation+ pyrolysis | Sandwich-like | 527.5/0.1/100 th 226/0.5/400 th | 206/5 157/10 | 72.6%/0.1 | PIB | [138]/2021 |

**Table 2.** Review of the properties of CoTe$_x$ electrode materials.

| Sample | Synthesis Method | Structure | Cycle Performance $^x$ | Rate Performance (mAh g$^{-1}$, A g$^{-1}$) | Initial Coulombic Efficiency (%, A g$^{-1}$) | Battery Type | Ref./Year |
|---|---|---|---|---|---|---|---|
| CoTe$_2$–C | Annealing | Polyhedron | 500/0.1/200 th 480/1/200 th | 386/3 | 70.52%/0.1 56.07%/0.1 | LIB SIB | [52]/2020 |
| CoTe$_2$@3DG | Annealing | Polyhedron + rGO | 191/0.05/350 th 103/1/4500 th | 169/2 | 66.7%/0.05 | SIB | [139]/2023 |
| CoTe$_2$@NMCNFs | Electrospinning + annealing | Carbon fiber network | 261.2/0.2/300 th 143.5/2/4000 th | 152.4/10 | 57.1%/0.2 | SIB | [47]/2021 |
| CoTe$_2$@3DPNC | Annealing | Dual-type carbon | 216.5/0.2/200 th | 164.2/5 | 87.6%/0.2 | SIB | [140]/2022 |
| o–P–CoTe$_2$/MXene | Solvothermal annealing | MXene | 373.7/0.2/200 th 232.3/2/2000 th | 168.2/20 | 72.9%/0.05 | PIB | [38]/2021 |
| CTNRs/rGO | One-pot solvothermal | Nanorods/rGO | 306/0.05/100 th | 176/2 | 56%/0.05 | SIB | [141]/2019 |
| CoTe$_2$–C | Spray pyrolysis | Microsphere | 295.8/0.2/100 th | 163.7/2 | 69.2%/0.2 | PIB | [142]/2020 |
| Cu–Co$_{1-x}$Se$_2$@NC | Chemical etching tellurization | Hollow nanoboxes | 796/1/800 th | ~400/5 | ~80%/1 | LIB | [71]/2022 |
| CoTe$_2$/G | One-pot solvothermal | Nanosheets | 356/0.05/100 th | 246/5 | 78.1%/0.05 | SIB | [42]/2018 |
| ZnTe/CoTe$_2$@NC | Annealing | Bimetallic heterostructure | 317.5/0.5/1000 th 254.5/2/2000 th 165.2/5/5000 th | 190.2/5 | 58.3%/0.1 | PIB | [40]/2022 |
| o/h–CoTe$_2$ | One-step hydrothermal | Submicron-sized rods | 807/0.12/200 th 438/0.6/400 th | 305.8/3 | 75.2%/0.12 | LIB | [93]/2023 |
| CoTe@NCD | Simultaneous pyrolysis–tellurium melt impregnation | Carbon dodecahedra | 300/0.1/100 th 200/0.1/100 th | 207/2 58/2 | 57%/0.1 50%/0.05 | SIB PIB | [129]/2022 |
| CoTe$_2$@NPC-NFs@NC | Electrospinning | N-doped porous carbon nanofibers | 409.1/0.05/50 th 198/0.5/600 th 120/2/1000 th | 148.9/2 | 57.2%/0.05 | PIB | [41]/2023 |

In general, cobalt selenide has been shown to be a high-energy anode for LIBs, SIBs, and PIBs, which greatly improves their electrochemical performance through independent electrodes without binders and with innovative heterojunction design. The reaction mechanism has also been studied by theoretical calculation and in situ techniques. As a potential electrode with high area capacity, cobalt telluride has many surprising applications in PIBs, and some electrode design methods that have proven excellent for cobalt selenide, such as heterojunctions and composite MXenes, have also been efficiently practiced for CoTe. However, compared with the metal oxides, sulfides, or selenides, the existing modification strategies are still insufficient. In addition, electrochemical performance can be enhanced by electrode synthesis and preparation. For example, studies on the compatibility of cobalt telluride with electrolytes and binders are still lacking. For practical applications, in future, more investigations and full battery evaluation should be emphasized.

## 6. Summary and Expectations

Cobalt selenide and cobalt telluride show promising applications in LIBs, SIBs, and PIBs due to their unique crystal structures, diverse structural designs, and high theoretical capacities. In this review, the progress of CoSe and CoTe was comprehensively summarized, not only focusing on the running mechanism and electrochemical performance, but also putting forward challenges to the synthesis strategies. The mechanism of Co-based chalcogenides in energy storage reactions is complex because it may have mechanisms for multilevel transformation. Often, conversion reactions are indispensable, which can lead to high weight capacities. However, disadvantages remain, such as large volume expansion and sluggish kinetics. In response to this series of problems, effective improvement strategies have been reported, including the design of nanocomposite carbon materials and the introduction of defects. Firstly, 0–3D conductive carbon nanocomposites can improve conductivity, create abundant active sites, and enhance electrolyte penetration. In addition, the introduced vacancies and heterogeneous interfaces can accelerate the conduction of ions and electrons. Tables 1 and 2 summarize the latest modified properties and synthesis methods, which can provide a reference point for innovative strategies for cobalt selenide and cobalt telluride materials. Nevertheless, the application of cobalt selenide and cobalt telluride anode materials in alkali-metal-ion batteries still requires extensive efforts. Therefore, we propose some prospective directions to promote further research to enable cobalt selenide and cobalt telluride to achieve higher energy storage efficiency.

### 6.1. Replenishing Existing Strategies

Heteroatom doping is an effective strategy to increase the numbers of ion storage locations and ion mobility channels, thereby facilitating faster electron transport. However, it is often reported that only one metallic element or non-metallic element (such as the common elements S and P) is selected to be mixed into cobalt selenide or cobalt telluride. At the same time, to improve the conductivity and active sites of $CoSe_x$ and $CoTe_x$, the selection of composite carbon materials has been limited to traditional carbon materials, such as rGO, dopamine, super-C, or metal–organic frameworks. In addition, many reports testify that MXenes have the advantages of large interlayer spacing, good conductivity, and heteroatom doping. In summary, the new strategy of multi-element doping and synthesis and the novel MXene composite method can improve the performance of cobalt telluride and cobalt selenide as anodes for alkali-ion batteries.

### 6.2. Expanded CBs' Anode Material Synthesis Method

Compared to $CoO_x$ and $CoS_x$, the available modification strategies are still limited. Currently, constructed multilayer, porous, or three-dimensional mesh structures can provide short transmission channels and large specific surface areas with active sites. This helps to improve material toughness and electrolyte contact, thus improving long-cycle performance. Relevant synthesis methods have been reported for $CoSe_x$, but less often for $CoTe_x$. Based on this, we should further explore the latter's electrochemical performance in LIBs/SIBs/PIBs. For example, three-dimensional nanonetworks of $CoTe_x$ based on hydrogels may be a good synthesis method. Furthermore, binder-free electrodes for $CoTe_x$ are rarely reported. Independent electrodes can be prepared via carbon cloth chemical deposition and electrospinning to avoid the obstruction of poor-conducting binders and low-capacity conductive agents, thus saving resources, reducing impurities, and improving the energy density.

**Author Contributions:** Writing—review and editing, Y.Z.; supervision, Z.S. and D.Q.; resources, D.H. and L.N. All authors have read and agreed to the published version of the manuscript.

**Funding:** This work was financially supported by the National Natural Science Foundation of China (22204028, 22104021, 22204159), Young Talent Support Project of Guangzhou Association for Science and Technology (QT-2023-003), Guangdong Basic and Applied Basic Research Fund Project (2022A1515110451), Guangzhou University Graduate Student Innovation Ability Cultivation Fund-

ing Program (2022GDJC-M06), and Science and Technology Projects in Guangzhou (202201010245, 2023A03J0029).

**Institutional Review Board Statement:** Not applicable.

**Informed Consent Statement:** Not applicable.

**Data Availability Statement:** Not applicable.

**Conflicts of Interest:** The authors declare no conflict of interest.

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
