# Peer review of "Recent Advances in CoSex and CoTex Anodes for Alkali-ion Batteries"

_coatings, doi:10.3390/coatings13091588_

Round 1
Reviewer 1 Report
Comments:
1. Authors must add decent 1080-pixel images most of the images seems to be blurred or out of pixel.
2. In the introduction it would be good if the authors include much technical information about why TMC's are being widely used and how they can revolutionalize batteries with their properties kind off.
3. Also why Co-based materials were widely used physiochemical properties inclusion will be of a great deal.
4. In 3.2 authors discussed the structural design it would be great if they concentrate more on how diffusion kinetics works and how that impacts conductivity and compatibility.
5. Every material has its limitations like MXene, MOF, and carbon-based it would be good if the authors include those limitations and explain how researchers are trying to override them.
6. Cite these references and discuss more
Jana, J., Sharma, T. S. K., Chung, J. S., Choi, W. M., & Hur, S. H. (2023). The role of surface carbide/oxide heterojunction in electrocatalytic behavior of 3D-nanonest supported Feimoj Composites. Journal of Alloys and Compounds, 946, 169395.
Catalysts 2020, 10(5), 495; https://doi.org/10.3390/catal10050495
Nguyen, V. H., Nguyen, B. S., Hu, C., Nguyen, C. C., Nguyen, D. L. T., Nguyen Dinh, M. T., ... & Le, Q. V. (2020). Novel architecture titanium carbide (Ti3C2Tx) MXene cocatalysts toward photocatalytic hydrogen production: a mini-review. Nanomaterials, 10(4), 602.
English writing is a bit moderate and needs revision through a native speaker.
Author Response
We would like to thank the reviewers for their valuable comments, which help us to improve the manuscript. The relevant parts of the manuscript were modified according to the reviewers’ comments. Below is our point-by-point response to each of the comments.

Reviewer 2 Report
The authors have written a thorough review please accept the manuscript with the below suggested minor changes:
1) The images in Fig.1 are not legible can it be made into a table or only small info can be provided in images?
2) Fig.5 image looks blurried, can authors paste high resolution mage of the same?
Author Response

(The authors gave the same response as above.)

Reviewer 3 Report
This review provides a comprehensive and timely analysis of CeSex/CoTex electrodes for battery applications. However, we would like to offer the following comments for consideration:
1) Figure 1: It would enhance clarity if references were included alongside corresponding years/illustrations in Figure 1.
2) Electrochemical Reaction Mechanisms: The authors are encouraged to incorporate and elaborate upon the amorphization mechanism of these electrodes within the discussion of electrochemical reactions.
3) Figure Presentation: Some figures appear overcrowded and contain low-quality images, potentially hindering reader comprehension. We suggest rearranging and improving figure quality for better readability.
4) Metal Doping Strategy: When discussing the metal doping strategy, consider exploring the electrochemical activity of the dopant and its impact on capacity contribution. Additionally, discuss how this affects the electrochemical mechanism compared to the pristine counterpart.
5) Carbon-Based Electrodes: In the section on carbon-based electrodes (3.2.2), consider incorporating a discussion of their capacity contribution alongside transition metal dichalcogenides (TMDs).
6) Electrochemical Reaction Dynamics: Provide a more detailed discussion of electrochemical reaction dynamics, such as capacitive contributions, especially in the context of TMDs.
7) Ragone Plot: Including a Ragone plot illustrating representative outstanding electrode performance would be beneficial for readers to visualize the energy density and power density trade-offs.
8) Electrolyte Configuration and Additives: Consider adding a section or commentary on electrolyte configurations and additives that enhance the performance of TMD electrodes, providing insights into their influence on battery performance.
9) Full Cell Applications: Considering the availability of reports in the literature, it would be highly valuable to include insightful discussion of full cell applications of CeSex/CoTex electrodes.
Author Response

(The authors gave the same response as above.)

Reviewer 4 Report
In this paper, the authors reviewed papers of the recent advances of the synthesis and evaluation of cobalt selenide for Li, Na, and K ion battery anodes. The manuscript contains an overview of the synthesis of cobalt selenide-based electrode materials with various compositions and structures. The manuscript carefully summarized and described the related research in this field. However, the manuscript still has small mistakes. The authors should consider the following comments and revise the manuscript before publishing the manuscript on “coatings”.
1. In abstract, it is better to add a description of some remarkable results, such as actual value or materials, to attract the attention of readers.
2. In section 3.2, page 15, sub- title number should be “3.2.3” for MOFs-based electrode section.
3. In section 3.2, page 15, MOFs-based electrode section, it is better to add a figure with a schematic model of MOF.
There are some mistakes in the text and figure mentioned following.
1. In abstract, page 1, line 10, “~ Cobalt selenide ~“ should be “~ cobalt selenide ~ “.
2. In section 2, page 4, “~ at 1.42 V, 1.27 V, and 0.63 V~“ should be “~ at 1.42, 1.27, and 0.63 V ~ “.
3. In section 2 and 3, the authors use different style notation for figure, such as figure 3j (bold style), figure 3k, and Figure 3l. Please unify the style.
4. In section 3.1.3, page 12, line 368, “the more induced ~“ should be “The more induced ~ “. And please check the English writing again before re-submission.
Author Response

(The authors gave the same response as above.)
